# Persuasive Effects of Message Framing and Narrative Format on Promoting COVID-19 Vaccination: A Study on Chinese College Students

**DOI:** 10.3390/ijerph18189485

**Published:** 2021-09-08

**Authors:** WeiMing Ye, Qian Li, Shubin Yu

**Affiliations:** 1HSBC Business School, Peking University, Shenzhen 100871, China; yewm@phbs.pku.edu.cn; 2Walter Cronkite School of Journalism and Mass Communication, Arizona State University, Tempe, AZ 85281, USA; qianli11@asu.edu

**Keywords:** COVID-19, message framing, narrative framing, health belief model, health behavior, vaccination

## Abstract

During a public health crisis, the provision and dissemination of health-related information are important for the relevant authorities to keep the public informed. By using different types of message framing, the authorities can effectively guide and persuade people to adopt health-related behaviors (such as vaccination). In this study, a web-based experiment using a 2 × 2 (message framing: gain framing versus loss framing) × (message presentation: narrative versus non-narrative) design was conducted to investigate the effects of different message frames on vaccination promotion. In total, 298 college students were recruited to participate in this study. The results suggest that, for message framing, loss-framed (vs. gain-framed) messages lead to higher intentions to get vaccinated. Furthermore, compared with non-narrative messages, narrative messages are more persuasive in promoting vaccination behavior. However, the interaction effect between gain–loss message framing and narrative framing is not significant. Additionally, perceived severity, perceived benefits, and perceived costs mediate the effect of narrative framing on behavioral intentions. In other words, compared with non-narrative messages, narrative messages lead to higher levels of perceived severity and perceived benefits, and a lower level of perceived costs, which in turn increase intentions to get vaccinated. This paper provides insightful implications for both researchers and practitioners.

## 1. Introduction

During a public health crisis, the provision and dissemination of health-related information are important for the relevant authorities to keep the public informed. By using different types of message framing, the authorities can effectively guide and persuade people to adopt health-related behaviors (such as vaccination).

Previous studies in the field of health communication focused more on analyzing the effectiveness of specific information contexts and less on the relationships between message framing and behavioral intentions [1,2,3,4,5]. The way people process health-related information is not completely rational [6,7]; this indicates that the sole examination of media presentations cannot precisely measure the real persuasive effects of messages [8,9]. People’s decision-making preferences are also affected by how information is presented [10,11,12]. According to framing effect theory, different presentations of health-related information can affect individuals’ decision-making preferences. Therefore, grasping the process of interaction between information and people and designing effective information to influence people’s decision-making processes can produce a positive impact when communicating during a public health crisis.

Due to the asymmetries between people’s responses to and preferences for information expressions and their different attitudes toward various options during the decision-making process [13,14,15,16], the effects of gain framing and loss framing have been primarily discussed and compared in previous studies. The research proposed that whether the information was presented with benefits or risks would have significant and different impacts on people’s behavioral preferences [16]. Under the definition of gain–loss framing effect, both “benefit” and “risk” are expressed as a subjective view and personal feeling of possible or assumed consequences. Specifically, health information with the gain frame will focus on defining the gains obtained by people from accepting a specific behavior. The information with loss frame, on the other hand, will highlight the risks associated with the rejection of such a health behavior [17,18]. Then, gain- and loss-framed messages have emerged as the essential tool to examine the framing effect in health communication studies [16,17,18,19,20,21,22,23,24,25,26]. Many studies also found that more attention should be paid to research contexts when examining the effectiveness of message frameworks, and combining framed messages with specific contexts is a necessary aspect to be considered [19,20,21,22,23,24,25,26].

Moreover, different types of health behaviors have also been introduced as moderators in analyzing the relationships between the framing effect and behavioral intentions [17,18]. Specifically, loss-framed messages have proven to be more persuasive in encouraging detection behaviors [19,20,21]; conversely, messages presented in gains are more persuasive in encouraging prevention behaviors [22,23,24]. Thus, gain framing may be more effective in promoting vaccination, which has been considered one type of prevention behavior [23,24,25]. However, regarding the global COVID-19 pandemic that continues to influence daily life significantly worldwide, it is unknown whether a gain-framed message will be more effective than a loss-framed message in promoting vaccination against COVID-19.

According to exemplification theory, many studies have argued that people pay more attention to information represented with vivid and lucid cases, and, compared with messages expressed with statistical descriptions, those expressed with anecdotes play a greater role in persuasion [26,27,28,29]. The explorations and findings related to exemplification theory have also contributed to the study of the relationships between narratives and persuasive effects. Narratives are defined as one type of message format associated with a series of events and characters, and compared with non-narratives, narratives tell stories from the first-person perspective [29,30]. Many empirical studies have proven that narrative messages, compared with non-narrative messages, have a greater persuasive effect on promoting health behaviors [31,32,33]. In this case, the fictional and fascinating stories presented in narrative messages play a good role in transporting health information to their audiences by providing a sense of familiarity and imaginability and will largely reduce people’s perceptions of fear and uncertainty [33,34]. Based on this, it is also meaningful to examine the effectiveness of narrative messages in the promotion of COVID-19 vaccination.

People’s decisions are closely related to various psychological factors, including cognitions, emotions, attitudes, and intentions [35,36,37,38]. Some theoretical frameworks, including the Elaboration Likelihood Model (ELM) and the Health Belief Model (HBM), have been used to analyze and predict people’s health behaviors [39,40,41]. The ELM provides a general framework for organizing, categorizing, and understanding the fundamental processes underlying the effectiveness of persuasive communications [39]. As a critical variable in the ELM model, issue involvement is used to measure the importance or relevance of the information to individuals. Thus, some studies combined the theory of the framing effect with the ELM and introduced issue involvement as a mediator to analyze the framing effect on people’s behavioral intentions [39,42]. The HBM has always been seen as one of the most widely used mainstream theoretical frameworks. It is also the earliest theoretical model for exploring people’s attitudes and individual decision preferences. The HBM asserts that individuals’ attitudes and intention to adopt health-related behaviors depend on their health beliefs [40,41,43]. The HBM proposes that people’s intentions are caused by their perceived threats of specific diseases and their evaluation of the recommended preventive measures [41,43]. Specifically, four main health beliefs, namely, perceived severity, perceived susceptibility, perceived benefits, and perceived costs, have been extensively discussed [44,45,46,47,48]. Moreover, previous studies have also examined and proven the mediating effects of health beliefs on message framing [49,50,51]. Conversely, it has also been proposed that health beliefs significantly mediate the interaction between framed messages and behavioral intentions.

According to the latest Ipsos survey conducted by the World Economic Forum, among more than 18,000 adults from 15 countries, about 73% agreed to get vaccinated against COVID-19; however, perhaps due to concerns about possible side effects resulting from the short clinical trials, 27% disagreed. As a result, the analysis of people’s attitudes and intentions toward COVID-19 vaccination is quite meaningful. Among different groups of people, many studies have focused on understanding the persuasive effects of messages in the promotion of the vaccination behavior of young adults. Specifically, a variety of studies have examined the effects of framing on promoting undergraduate students’ intentions to get the HPV vaccine [35,52,53]. For COVID-19 vaccination, increasingly more vaccines are being approved for young adults. However, promoting COVID-19 vaccination among young adults may be challenging, as they believe they are less at risk compared with older adults. Persuading young adults to get vaccinated is meaningful for the vaccination campaign as a whole and is also the key to achieving herd immunity. In addition, according to the latest report released by China’s Ministry of Education, as of 2020, the total enrollment in higher education in China was 41.83 million, with an enrollment rate of 54.4%. Because college students are considered important human resources for social development, their health conditions are associated with the future of the entire nation. Furthermore, the determination of how to improve the persuasiveness of health information to affect college students’ behavioral intentions via the manipulation of different message strategies helps to achieve health education on a larger scale [54]. Thus, the examination of the effect of framing on the promotion of health behavior intentions makes both academic and practical contributions. Therefore, the present research aims to explore how different message framings affect their persuasiveness in promoting COVID-19 vaccination for young adults.

To inform and examine the effectiveness of framed messages, this research seeks to understand how intentions to get vaccinated are influenced by message framing and message presentation. Furthermore, this research investigates the mediating effects of health beliefs on the relationship between framing and intentions. Specifically, this study aims to answer the following research questions. (1) Do gain-framed and loss-framed messages have different persuasive effects on COVID-19 vaccination intention? (2) Do narrative messages and non-narrative messages have different persuasive effects on COVID-19 vaccination intention? (3) Do message frames interact with narrative framing to influence COVID-19 vaccination intention? (4) How do narrative and non-narrative messages affect respondents’ health beliefs? (5) Are the effects on vaccination intention mediated by health beliefs?

## 2. Materials and Methods

### 2.1. Study Design and Participants

A web-based experiment using a 2 × 2 (message framing: gain framing versus loss framing) × (message presentation: narrative versus non-narrative) design was conducted. College students were recruited from an online panel run by Wenjuanxing (available online: https://www.wjx.cn/ (accessed on 19 March 2021)), which is like Amazon’s Mechanical Turk and Prolific. Students who have never received any COVID-19 vaccine were included in this study. The study was conducted according to the guidelines of the Declaration of Helsinki, and approved by the Discipline and Ethics Committee of Peking University HSBC Business School (PHBS0401, 21 April 2021). Following Simmons, Nelson, and Simonsohn’s recommendations [55], we planned to target at least 50 per cell in online studies. As the final sample, 298 students (*M*age = 21.44; 46.3% female; 63.7% undergraduate students and 36.3% graduate students (including 0.7% Ph.D. students) were recruited to participate in this study. The participants were invited to browse a web page, where they were shown a brief introduction and asked to sign an informed consent form. Once they agreed to participate, they were randomly assigned to one of the four experimental conditions. The participants were primarily asked to report their basic knowledge about COVID-19 and the associated vaccines. Then, the participants were presented with one piece of health messaging promoting COVID-19 vaccination. The message framing was manipulated; the message was either gain-framed or loss-framed. The message presentation was also manipulated; the message was presented in either a narrative or non-narrative format. After reading the message stimuli, the participants were asked to answer a series of questions. A set of Likert scales was used to measure several variables, including the behavioral intention to get the COVID-19 vaccine and health beliefs toward vaccination. Participants were finally asked to report demographic information, including their age, gender, location, education level, and income level.

The final sample was composed of 298 college students. The participants in this experiment lived in 29 provinces in China, accounting for 85% of the total number of provinces in the country. Based on the ratio of the geographical distribution of participants, the participants were evenly distributed between east-west and north-south. Thus, the participants could represent the basic characteristics of Chinese college students, and the bias caused by regions could be effectively avoided. A brief survey was conducted to test the students’ basic knowledge and cognition related to both COVID-19 and the COVID-19 vaccines. This survey found that no participants in the selected sample had been infected with COVID-19, 84.3% said that they had previously heard of the COVID-19 vaccine, and 15.7% said that they had not previously heard of it. Most participants believed that they had a good understanding of the mechanism of the COVID-19 vaccine and believed that accepting vaccination is quite beneficial to human health; however, only half of the Chinese population believe that their daily life is closely related to the COVID-19 vaccine.

### 2.2. Message Stimuli

The health messages were presented as newsletters, which provided basic information regarding COVID-19 and the COVID-19 vaccines. Then, two features of the message were designed, including message framing and message presentation. Following prior studies [56,57], the gain-framed messages conveyed the benefits of getting vaccinated, while the loss-framed messages focused on the potential detriments of not getting vaccinated. In addition, following previous research [35,54], the narrative messages were manipulated by presenting a personal story from the first-person perspective. In the non-narrative messages, no specific characters were involved, and the messages conveyed an objective conclusion. The message stimuli are reported in Table 1.

### 2.3. Measurements

Unless indicated otherwise, the responses to items were given on 5-point Likert scales. The Likert scale ranging from 1 (strongly disagree) to 5 (strongly agree) as endpoints and the average mean values of related items were considered as indexes for the related variables. Before measuring all related variables, this study conducted the manipulation checks to examine the efficiency and reliability of the design of message stimuli. This study then measured the dependent variable, that is, intention to get the COVID-19 vaccine. Then, the present study continued to measure four main health beliefs as mediators, which include perceived severity, perceived susceptibility, perceived benefits, and perceived costs. Finally, this study measured demographical variables, including the participants’ age, gender, education level, and Internet literacy.

#### 2.3.1. Manipulation Checks

The present study conducted a set of manipulation checks of the framed messages before measuring all related variables. The purpose of this is to test the effectiveness and accuracy of the message designs. The manipulation checks contained two dimensions of measurement: one is to measure how much participants think this is a piece of positive or negative message, and the other is to measure how much participants think this message is a narrative or non-narrative one. This study asked participants to complete the scoring of the following questions based on their feelings after reading the assigned health message, and the Likert scale ranging from 1 (strongly disagree) to 5 (strongly agree) was also used in the questionnaire. Specifically, six items were included: “This information emphasizes the benefits of getting COVID-19 vaccination”; “This information emphasizes the risk of not getting COVID-19 vaccination”; “This information emphasizes that getting COVID-19 vaccination will bring you positive effects”; “This information emphasizes that getting COVID-19 vaccination will bring you negative effects”; “This information is written in a person’s narrative”; “This information is written in an objective style”.

#### 2.3.2. Intention to Get the COVID-19 Vaccine

The measurement of the intention to get the COVID-19 vaccine was adapted from past studies [54] that measured people’s intentions to get a vaccination in both the short term and the long term. Participants were asked to respond to three questions (e.g., “How likely would you be to get the COVID-19 vaccine sometime soon?”; Cronbach’s α = 0.77, *M* = 3.71, *SD* = 1.14). Higher scores indicated participants’ stronger intention to get vaccinated.

#### 2.3.3. Health Beliefs

Scales adapted from previous studies [57,58] were used to measure the four specific health beliefs. Three items assessed perceived susceptibility by measuring the perception of risks associated with COVID-19 and the possibility of infection (e.g., “I may get COVID-19”; Cronbach’s α = 0.89, *M* = 3.26, *SD* = 1.41). Three items assessed perceived severity by measuring the perception of the negative consequences caused by COVID-19 (e.g., “I believe that COVID-19 will result in severe health problems”; Cronbach’s α = 0.71, *M* = 3.72, *SD* = 1.07). Another three questions were posed to test perceived benefits by measuring the evaluation of vaccine efficacy in preventing COVID-19 (e.g., “I believe if I get the COVID-19 vaccine, I will be less likely to get COVID-19”; Cronbach’s α = 0.76, *M* = 3.74, *SD* = 1.15). Furthermore, three items were used to assess perceived costs by measuring the perception of barriers from getting vaccinated (i.e., “I worry about the short-term side effects of the COVID-19 vaccine”; “I worry that the COVID-19 vaccine might negatively affect my body”; “I worry that the COVID-19 vaccine might have unknown long-term side effects”; Cronbach’s α = 0.83, *M* = 2.39, *SD* = 1.26). Higher scores indicated stronger health beliefs.

#### 2.3.4. Control Variables

The choice of college students as research participants may have led to greater similarity between groups. The age, gender, education level, and Internet literacy of the participants were therefore included in the analyses as control variables.

### 2.4. Data Analysis Strategies

To test the research questions, series of analyses of variance (ANOVAs) and analyses of mediation were conducted. Via ANOVA, the differences between experimental groups were compared while controlling for demographic variables. The main effects of message framing and message presentation were first examined, after which the interaction effect between message framing and message presentation was tested. The PROCESS macro (version 3.5, Andrew F. Hayes, Columbus, OH, USA) for SPSS 27 (IBM, Armonk, NY, USA) was used for mediation analysis (Model 4) because it adopts a bootstrap method to estimate the mediating effect [59]. Via mediation analyses, the mediating effects of the four health beliefs on the relationship between narrative message presentation and the intention to get the COVID-19 vaccine were examined.

## 3. Results

### 3.1. Manipulation Checks Results

Manipulation checks were conducted on message framing (gain framing versus loss framing) and message format (narrative versus non-narrative) using a series of independent t-tests. The results showed that participants in the gain-framed message condition perceived the message to focus more on expressing the positive information related to vaccination (*M* = 3.966, *SD* = 0.519) as compared to those in the loss-framed message condition (*M* = 3.3226, *SD* = 1.2709), *t* = 5.647, *p* < 0.001. Moreover, participants in the narrative message condition were more likely to perceive the message to be presented from the personal perspective (*M* = 4.1769, *SD* = 0.7375) than were those in the non-narrative message condition (*M* = 2.0204, *SD* = 0.8148), *t* = 23.7902, *p* < 0.001. Therefore, both manipulations were successful.

### 3.2. Main Effects

To answer RQ1, whether gain-framed and loss-framed messages have different persuasive effects on COVID-19 vaccination intention, the results show that for message framing, the loss-framed message, compared with the gain-framed message, promoted the intention to get the COVID-19 vaccine. A one-way ANOVA was conducted on the COVID-19 vaccination intention, and message framing was entered as the independent variable. As shown in Table 2, the results indicate that the main effects of message framing on vaccination intention were statistically significant, *F* (1, 296) = 9.855, *p* = 0.002. Specifically, the loss-framed message led to stronger behavioral intention (*M*loss-framed = 3.895 versus *M*gain-framed = 3.563). Furthermore, another one-way ANOVA was performed to answer RQ2, which is whether narrative and non-narrative messages have different persuasive effects. The message format was introduced as the independent variable, and vaccination intention was considered as the dependent variable. The results indicate that the main effects of the message format on vaccination intention were statistically significant, *F* (1, 296) = 11.334, *p* = 0.001. Specifically, the narrative message promoted the intention to get the COVID-19 vaccine (*M*narrative = 3.908 versus *M*non-narrative = 3.552). The interaction between the loss-gain framing and narrative framing was examined, and the interaction effect on vaccination intention was not found to be statistically significant. Thus, for RQ3, whether gain–loss framing interacts with narrative framing on COVID-19 vaccination intention, the interaction was not statistically significant.

### 3.3. Mediation Effects

RQ4 tried to investigate whether and how narrative and non-narrative messages affect college students’ health beliefs, including perceived susceptibility, perceived severity, perceived benefits, and perceived costs. To address this question, an analysis of mediation was conducted by using the PROCESS macro in SPSS, and the model 4 was selected. Using a bootstrap method, this study selected a sample size of 5000. Narrative message presentation was entered as the independent variable, vaccination intention was introduced as the dependent variable, and health beliefs were included as mediators. As shown in Table 3, the results indicate that the mediation effects of health beliefs on narrative framing were partially significant. When testing the mediation effects of perceived severity, perceived benefits, and perceived costs, their confidence intervals of the bootstrap did not include zero, which means that those three variables had significant mediating effects on narrative framing. Specifically, the mediating effects of perceived severity (BootLLCI = 0.0186, BootULCI = 0.0868), perceived benefits (BootLLCI = 0.0137, BootULCI = 0.1030), and perceived costs *(*BootLLCI = 0.0046, BootULCI = 0.0660) were statistically significant in the 95% bias-corrected bootstrap confidence interval. Therefore, perceived severity, perceived benefits, and perceived costs mediate the relationship between narrative framing and behavioral intention. RQ5 then investigated whether the health beliefs mediate the relationship between narrative framing and intentions to get the vaccination. To address this research question, the coefficients of the mediating models were examined to analyze the directions of the mediating effects and the results were shown in Figure 1. Specifically, compared with non-narrative messages, narrative messages led to a higher level of the perceived severity of COVID-19 and the perceived benefits of the COVID-19 vaccine. Furthermore, compared with the non-narrative messages, the narrative messages led to a lower level of perceived costs. In conclusion, narrative messages were found to lead to higher levels of perceived severity and perceived benefits, while they led to a lower level of perceived costs, and therefore were ultimately found to promote the intention to get the COVID-19 vaccine.

## 4. Discussion

This present research is one of the first few studies to investigate the effects of message framing and narrative message presentation on promoting COVID-19 vaccination. Specifically, the persuasiveness of gain-framed versus loss-framed messages was compared, as was the persuasiveness of narrative versus non-narrative messages. Because the research targets were Chinese college students, it is difficult to compare the results of the present research with those of previous studies due to the lack of surveys conducted among the same group. This research, however, revealed some notable findings in the prediction of the intention to get the COVID-19 vaccine.

A central thesis of this research is that vaccination, unlike other preventative health-related behaviors, is associated with higher risks due to side effects and other safety concerns. Moreover, under the influence of public opinion, the anxiety of the public will spread, causing a large portion of the population to be reluctant to get vaccinated even if they believe that vaccination is a beneficial behavior. Drawing upon prospect theory [16,53,60], people will prefer less risky behavior when those risks are expressed salient. Because people will exhibit greater aversion to the risks caused by vaccination, loss-framed messages are more likely to enhance their intention to get the COVID-19 vaccine. The results of the controlled experiment were found to be largely consistent with the research questions, and the findings are consistent with the results of some previous studies conducted in other contexts [33,34,53,54].

Similarly, this research also posited that narrative messages are more persuasive than non-narrative messages to promote COVID-19 vaccination behavior. Narrative descriptions associated with fictional and fascinating stories will benefit people to build a more specific and concrete understanding of the issue [28,29,30,60]. Furthermore, compared with didactic and objective arguments, messages in narrative format provide people with more familiarity and imaginability, and will therefore be more persuasive in promoting behavioral intentions. Consistent with the prediction, the results showed that narrative messages, both gain-framed and loss-framed, are more persuasive than non-narrative messages in promoting vaccination. These findings are consistent with those of previous studies that have suggested the greater effectiveness of narrative messages in promoting vaccination [33,34,61].

In addition to examining the effects of message framing and presentation, this research examined the mediating roles of various health beliefs on the interaction between narrative framing and vaccination intention. Drawing upon the HBM, people’s decision-making processes can be influenced by their evaluations of potential threats and efficacy [41,42,62]; thus, many studies have introduced health beliefs into a framing effect model to examine their mediating effects [34,48,49]. In the present research, it was found that health beliefs significantly mediate framing effects; this is consistent with the findings of some previous studies [33,62,63,64], even though they were focused on other types of health behaviors or other framing effects. Specifically, narrative (versus non-narrative) messages will lead to higher levels of perceived severity and perceived benefits and will simultaneously lead to a lower level of perceived costs; thus, they will promote the intention to get the COVID-19 vaccine. Messages conveyed in narrative format will enhance people’s perceived threats of COVID-19 and the perceived efficacy of the vaccines. Furthermore, messages presented as narratives will help people to better understand and avoid the well-demonstrated risks, and, conversely, will lessen their concerns about vaccine side effects and other safety barriers.

Of course, all the conclusions of this research must be evaluated in consideration of several limitations. This research failed to introduce discrete emotions as mediators in analyzing the interactions between framing effects and intentions. Although the research fully measured and investigated people’s health beliefs, people’s emotions, including fear, sadness, guilt, and relief, can also lead to different behavioral responses [30,62,65,66,67,68,69,70]. Future research may seek to examine the mediating roles of emotions on framing effects, especially the mediating role of fear. People with fear tend to retreat from loss-framed stimuli and will avoid any behavior to address them [65,71,72]. In other words, people can be too afraid to make any possible attempt, even if they clearly know that such an attempt is beneficial to them. Additionally, future research can introduce the Extended Parallel Process Model (EPPM) into the discussion. Compared with the HBM, the EPPM can better measure people’s perceived threats and efficacy and can therefore be used to comprehensively examine the effect of framing on the persuasion of people’s behavioral intentions [56]. Finally, the present study did not examine the roles of individual affective and cognitive orientations on promoting behavioral intentions. This study merely focused on analyzing the persuasiveness of messages with different expressions. However, people’s affective and cognitive attitudes should also be introduced into the model. According to the theory of matching effect, if the framed message matches an individual’s affective and cognitive orientations, it will enhance the effectiveness of persuasion [73]. In other words, matches between people’s psychological states and the message will make it more persuasive [74,75,76]. Those related theories do give great inspiration for future research. According to the findings in this study, loss framing, compared with gain framing, is more persuasive. Additionally, narratives also have a more significant persuasive effect compared with non-narratives. However, this study did not take affective orientation and cognitive orientations into consideration. What type of framed messages can significantly match individual’s psychological states and will better trigger their behavioral intentions? In future research, it may be necessary to add more variables to measure the dimensions of personal psychological characteristics to better measure the influence of emotion and cognition orientations on persuasiveness.

## 5. Conclusions

In conclusion, the results of this study emphasize that the persuasive effects of various message expressions are significantly different. First, the difference between gain framing and loss framing in the promotion of vaccination intention was proven to be significant. Specifically, loss-framed messages are more persuasive than gain-framed messages in promoting COVID-19 vaccination. Because vaccination is one type of health behavior associated with risk, according to prospect theory, loss-framed messages are more persuasive. Second, the difference between narrative and non-narrative messages in encouraging vaccination was also proven to be significant. In other words, narrative messages are more effective than messages presented in a non-narrative format, as narrative messages allow people to better understand the potential risks of rejecting vaccination against COVID-19 or the benefits of getting the vaccine. Furthermore, narrative descriptions will enhance people’s familiarity with the framed message and more strongly trigger their intention to get the COVID-19 vaccine. Moreover, the mediating effects of health beliefs, including perceived severity, perceived benefits, and perceived costs, were proven to be significant. Specifically, messages conveyed in a narrative format will increase people’s perceptions of the severity of COVID-19 and the benefits obtained from getting the vaccine. Additionally, perceived costs will play a negative mediating role, i.e., narrative messages will result in a lower level of perceived costs and will make people more likely to get the COVID-19 vaccine.

## Figures and Tables

**Figure 1 ijerph-18-09485-f001:**
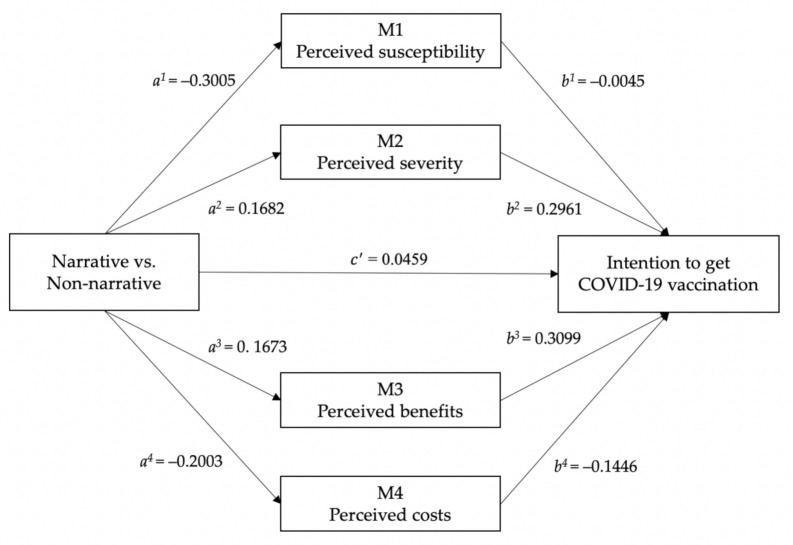
Mediating model on narrative framing.

**Table 1 ijerph-18-09485-t001:** Message stimuli in experiment conditions.

Gain-Framed	Loss-Framed
Non-narrative Scientific studies and clinical trials have shown that getting the COVID-19 vaccine will bring benefits to the human body. After being vaccinated, your body will produce protective antibodies, thereby developing immunity to the virus, which will greatly reduce your likelihood of contracting COVID-19 and protect your life from the epidemic. You will no longer have to worry about being quarantined or required to undergo nucleic acid amplification testing at any time. Getting vaccinated can also make you better able to protect those around you and greatly reduce your likelihood of infecting others, especially those who are more susceptible to COVID-19.	Non-narrative Scientific studies and clinical trials have shown that not getting the COVID-19 vaccine will bring harm to the human body. If you do not get vaccinated, your body will not be able to produce protective antibodies, and you will not be able to develop immunity to the virus, which will greatly increase the likelihood that you will be infected with COVID-19. If you don’t get vaccinated, you will be plagued by the epidemic. You will often worry about being quarantined or required to undergo nucleic acid amplification testing. If you don’t get vaccinated, you won’t be able to better protect those around you. You will easily spread the virus to those around you, especially those who are more susceptible to COVID-19.
Narrative Scientific studies and clinical trials have shown that getting the COVID-19 vaccine will bring benefits to the human body. The following is a self-report by Mr. Zhang, a vaccine volunteer from Wuhan: “I was in one of the first batches of volunteers to be vaccinated in Wuhan. The vaccination went smoothly and took only a few tens of seconds, just like a normal vaccine. No adverse reactions occurred in my body, and there was no redness or swelling at the injection site. Although I was quite worried about the severity of this epidemic, the vaccination made my body develop antibodies, which gave me immunity to the virus, so I am not afraid anymore. I really feel a sense of steadfastness that has been long gone! I feel that the health of myself and my family is completely guaranteed.”	Narrative Scientific studies and clinical trials have shown that not getting the COVID-19 vaccine will bring harm to the human body. The following is a self-report by Mr. Zhang, a person infected with COVID-19 from Wuhan: “I could have made an appointment for vaccination in Wuhan at the end of this year, but I didn’t go because I was worried about the potential risk, and the process was a little troublesome for me. Later, I felt soreness in my throat and had a fever a week later. Then, I went to the hospital for testing, and it was confirmed that I was infected with COVID-19. The doctor told me that if there is no vaccine and no antibodies to the virus are produced in the body, there is always a risk of infection. I really regretted that if I had been vaccinated earlier and got immunized, I wouldn’t be infected. So, don’t take any chances, and get vaccinated in time!”

**Table 2 ijerph-18-09485-t002:** Means and standard deviation related to research questions 1–2.

Dependent Variable: Intentions to Get the Vaccination								
Group	*n*	Mean	Standard Deviation	Standard Error Mean	95% Confidence Interval of the Difference		Minimum	Maximum
					Lower	Upper		
Gain	145	3.5629	1.0088	0.0838	3.3973	3.7284	1.0000	5.0000
Loss	153	3.8954	0.8142	0.0658	3.7654	4.0255	1.0000	5.0000
Total	298	3.7336	0.9276	0.0537	3.6289	3.8394	1.0000	5.0000
Non-Narrative	146	3.5521	1.0196	0.0844	3.3854	3.7189	1.0000	5.0000
Narrative	152	3.9079	0.7948	0.0645	3.7805	4.0353	1.0000	5.0000
Total	298	3.7336	0.9276	0.0537	3.6279	3.8394	1.0000	5.0000

**Table 3 ijerph-18-09485-t003:** Mediation effects related to research questions 4–5.

**Total effect of X on Y**	**Effect**	** *SD* **	** *t* **	** *p* **	**LLCI**	**ULCI**
	0.1779	0.0528	3.3667	0.0009	0.0739	0.2819
**Direct effect of X on Y**	**Effect**	** *SD* **	** *t* **	** *p* **	**LLCI**	**ULCI**
	0.0459	0.0448	1.0253	0.3061	–0.0422	0.1340
**Indirect effect of X on Y**	**Effect**	**Boot*SE***	**BootLLCI**	**BootULCI**		
Total	0.1320	0.0368	0.0625	0.2089		
M1	0.0013	0.0107	–0.0196	0.0228		
M2	0.0498	0.0174	0.0186	0.0868		
M3	0.0519	0.0233	0.0137	0.1030		
M4	0.0290	0.0161	0.0046	0.0660		

## Data Availability

The data that support the findings of this study are available from the corresponding author, upon reasonable request.

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
