# Peer review of "Persuasive Effects of Message Framing and Narrative Format on Promoting COVID-19 Vaccination: A Study on Chinese College Students"

_ijerph, 2021, doi:10.3390/ijerph18189485_

Round 1

Reviewer 1 Report

Dear Authors,

Congratulations on such an interesting and innovative study. The study is overall well-written and the research questions are well-presented, well-developed and well-referenced.

Still, there are some issues you should take into consideration in order to make the paper more comprehensive:

  • In the abstract, you should very briefly provide some details about the sample and method of research you used.
  • The article refers to several theories such as the Elaboration Likelihood Model, the Health Belief Model (HBM) and others. They may not be known to all the readers of the article. The suggestion is to describe them in a short paragraph, when you mention them in the article. 
  • Section 3.1 is entitled Demographic Characteristics of Participants. Do you have other demographic characteristics of the sample you have analyzed or just the ones you have mentioned in this paragraph? 
  • At section 3.3. Main Effects, the research questions are written as RQ1, RQ2... This makes the person reading the article return to the research questions to know exactly what it is about. The suggestion would be for the research questions to be explicitly mentioned in the analysis part. 
  • In the conclusion section, you have made this statement: “A central thesis of this research is that vaccination, unlike other preventative health related behaviors, is associated with higher risks due to side effects and other safety concerns. Moreover, under the influence of public opinion, the anxiety of the public will spread, causing a large portion of the population to be reluctant to get vaccinated even if they believe that vaccination is a beneficial behavior. Drawing upon prospect theory [16, 53], people will prefer less risky behavior when those risks are expressed salient. Because people will exhibit greater aversion to the risks caused by vaccination, loss-framed messages are more likely to enhance their intention to get the COVID-19 vaccine. ” Is this a conclusion based on the results obtained in the research that you have carried out? 

Kind regards,

The Reviewer

Author Response

Revision Notes - ijerph-1361225

Dear Editor,

Dear Reviewers,

Thank you for your constructive feedback and useful remarks for improving the manuscript. We have taken the suggestions for improvement at heart and we have revised our manuscript accordingly. Below, you may find a detailed overview of our adjustments according to the comments raised by the reviewers.

Kind regards,

The authors

Comments raised by Reviewer 1:

Reviewer #1: Dear Authors,

Congratulations on such an interesting and innovative study. The study is overall well-written and the research questions are well-presented, well-developed and well-referenced.

Thanks very much for your constructive feedback.

Still, there are some issues you should take into consideration in order to make the paper more comprehensive:

  • In the abstract, you should very briefly provide some details about the sample and method of research you used.

Thanks very much for the comment. We have provided detailed information about the sample and research method in the abstract.

“Abstract: During a public health crisis, the provision and dissemination of health-related information are important for the relevant authorities to keep the public informed. By using different types of message framing, the authorities can effectively guide and persuade people to adopt health-related behaviors (such as vaccination). In this study, a web-based experiment using a 2 × 2 (message framing: gain framing versus loss framing) × (message presentation: narrative versus non-narrative) design was conducted to investigate the effects of different message frames on vaccination promotion. In total, 298 college students were recruited to participate in this study. The results suggest that, for message framing, loss-framed (vs. gain-framed) messages lead to higher intentions to get vaccinated. Furthermore, compared with non-narrative messages, narrative messages are more persuasive in promoting vaccination behavior. However, the interaction effect between gain-loss message framing and narrative framing is not significant. Additionally, perceived severity, perceived benefits, and perceived costs mediate the effect of narrative framing on behavioral intentions. In other words, compared with non-narrative messages, narrative messages lead to higher levels of perceived severity and perceived benefits, and a lower level of perceived costs, which in turn increase intentions to get vaccinated. This paper provides insightful implications for both researchers and practitioners.

  • The article refers to several theories such as the Elaboration Likelihood Model, the Health Belief Model (HBM) and others. They may not be known to all the readers of the article. The suggestion is to describe them in a short paragraph, when you mention them in the article. 

Thanks for your comment. We have added a more detailed explanation of the two models.

“People’s decisions are closely related to various psychological factors, including cognitions, emotions, attitudes, and intentions [36-39]. Some theoretical frameworks, including the Elaboration Likelihood Model (ELM) and the Health Belief Model (HBM), have been used to analyze and predict people’s health behaviors [39-42]. The ELM provides a general framework for organizing, categorizing, and understanding the fundamental processes underlying the effectiveness of persuasive communications [39-40]. As a critical variable in the ELM model, issue involvement is used to measure the importance or relevance of the information to individuals. Thus, some studies combined the theory of the framing effect with the ELM and introduce issue involvement as a mediator to analyze the framing effect on people’s behavioral intentions [40,44]. The HBM has always been seen as one of the most widely used mainstream theoretical frameworks. It is also the earliest theoretical model for exploring people’s attitudes and individual decision preferences. The HBM asserts that individuals’ attitudes and intention to adopt health-related behaviors depend on their health beliefs [41,42]. The HBM proposes that people’s intentions are caused by their perceived threats of specific diseases and their evaluation of the recommended preventive measures [43]. Specifically, four main health beliefs, namely perceived severity, perceived susceptibility, perceived benefits, and perceived costs, have been extensively discussed [44-48]. Moreover, previous studies have also examined and proven the mediating effects of health beliefs on message framing [49-52]. Conversely, it has also been proposed that health beliefs significantly mediate the interaction between framed messages and behavioral intentions.”

  • Section 3.1 is entitled Demographic Characteristics of Participants. Do you have other demographic characteristics of the sample you have analyzed or just the ones you have mentioned in this paragraph? 

Thanks for your comments. Apart from gender, age, education, regions, and previous knowledge of Covid, we did not include other demographic characteristics of the sample. And in order to make the article easier to understand, we moved the part of demographic characteristics of participants from the results section to the method section.

“The final sample was composed of 298 college students. The participants in this experiment lived in 29 provinces in China, accounting for 85% of the total number of provinces in the country. Based on the ratio of the geographical distribution of participants, the participants were evenly distributed between east-west and north-south. Thus, the participants could represent the basic characteristics of Chinese college students, and the bias caused by regions could be effectively avoided. A brief survey was conducted to test the students’ basic knowledge and cognition related to both COVID-19 and the COVID-19 vaccines. This survey found that no participants in the selected sample had been infected with COVID-19, 84.3% said that they had previously heard of the COVID-19 vaccine, and 15.7% said that they had not previously heard of it. Most participants believed that they had a good understanding of the mechanism of the COVID-19 vaccine and believed that accepting vaccination is quite beneficial to human health; however, only half of the Chinese population believe that their daily life is closely related to the COVID-19 vaccine.”

  • At section 3.3. Main Effects, the research questions are written as RQ1, RQ2... This makes the person reading the article return to the research questions to know exactly what it is about. The suggestion would be for the research questions to be explicitly mentioned in the analysis part. 

Thanks for your comments. I added some detailed clarifications to those questions.

“To answer RQ1, whether gain-framed and loss-framed messages have different persuasive effects on COVID-19 vaccination intention, the results show that for message framing, the loss-framed message, compared with the gain-framed message, promoted the intention to get the COVID-19 vaccine.”

“Furthermore, another one-way ANOVA was performed to answer RQ2, which is whether narrative and non-narrative messages have different persuasive effects.”

“Thus, for RQ3, whether gain-loss framing interact with narrative framing on COVID-19 vaccination intention, the interaction was not statistically significant.”

“RQ4 tried to investigate whether and how narrative and non-narrative messages affect college students’ health beliefs, including perceived susceptibility, perceived severity, perceived benefits and perceived costs.”

“RQ5 then investigated whether the health beliefs mediate the relationship between narrative framing and intentions to get the vaccination.”

  • In the conclusion section, you have made this statement: “A central thesis of this research is that vaccination, unlike other preventative health related behaviors, is associated with higher risks due to side effects and other safety concerns. Moreover, under the influence of public opinion, the anxiety of the public will spread, causing a large portion of the population to be reluctant to get vaccinated even if they believe that vaccination is a beneficial behavior. Drawing upon prospect theory [16, 53], people will prefer less risky behavior when those risks are expressed salient. Because people will exhibit greater aversion to the risks caused by vaccination, loss-framed messages are more likely to enhance their intention to get the COVID-19 vaccine. ” Is this a conclusion based on the results obtained in the research that you have carried out? 

Thanks for your comments. This is not a conclusion based on the empirical analysis. However, it is a proposition based on the prospect theory. Although it is not our conclusion which can be directly explained by our statistical results, we believe it is relevant to include this part in the discussion section.

Reviewer 2 Report

The topic is really interesting, expecially during the COVID emergence. I hope my comments can help the authors in improving their manuscript

Please report more details about the literature on loss-gaining framing in persuasion, which is crucial for the present manuscript. The literature on this field is wide, as the authors themselves recognize, so they can give more details for readers, by adding also some examples of gain (vs loss) persuasive messages. Please justify better your rationale.

At the end of the Introduction please report your hypotheses and not only your research questions.

How did you establish the sample size?

Regarding the development of the persuasive messages, did you pre-test it before the study? How did you establish the length of each message?

Please report the manipulation check questions in the section about measurement.

Please report the order of presentation of scales, this is very important given that you are testing also the mediation model.

The demographic features of the sample cannot be considered as results, please move this section to the design and participants section.

In the description of Process MACRO for mediation analyses please report the bootstrap you have used (e.g., 1000, 2000, 5000). Further, explain to readers that the effect is significant whether the CIs do not include the zero value.

In mediation analyses results the authors wrote “RQ4 emphasized that perceived susceptibility, perceived severity, perceived benefits, and perceived costs will have mediating roles on the relationship between narrative framing and vaccination intention” In my opinion this is a hypothesis, rather than a research question. Research question is to investigate whether the perceived susceptibility (and the other variables) would have a mediating role on the relationship between narrative framing and vaccination intention. The same for RQ5

A wide literature in persuasion have showed that individual differences in affective and cognitive orientation can predict the effectiveness of narrative (vs informative) messages (the so called matching effect, Mayer & Tormala, 2010, “Think” Versus “Feel” Framing Effects in Persuasion https://doi.org/10.1177/0146167210362981 Aquino et al., 2020, Sense or sensibility? The neuro-functional basis of the structural matching effect in persuasion, https://doi.org/10.3758/s13415-020-00784-7 ).

It could be interesting to investigate the role of affective and cognitive orientation in the field the authors are investigating. Please add this idea for future research (and the suggested references) in Discussion, so you can extend your manuscript to affective-cognitive persuasive field.

Author Response

Revision Notes - ijerph-1361225

Dear Editor,

Dear Reviewers,

Thank you for your constructive feedback and useful remarks for improving the manuscript. We have taken the suggestions for improvement at heart and we have revised our manuscript accordingly. Below, you may find a detailed overview of our adjustments according to the comments raised by the reviewers.

Kind regards,

The authors

 Comments raised by Reviewer 2:

  • The topic is really interesting, expecially during the COVID emergence. I hope my comments can help the authors in improving their manuscript. Please report more details about the literature on loss-gaining framing in persuasion, which is crucial for the present manuscript. The literature on this field is wide, as the authors themselves recognize, so they can give more details for readers, by adding also some examples of gain (vs loss) persuasive messages. Please justify better your rationale.

Thank you for your insightful suggestions. We also believe it is important to include relevant literature on this field to better position this study. Therefore, we have added more information on gain-loss framing in the Introduction section.

“Due to the asymmetries between people’s responses to and preferences for information expressions and their different attitudes toward various options during the decision-making process [13-16], the effects of gain framing and loss framing have been primarily discussed and compared in previous studies. The research proposed that whether the information was presented with benefits or risks would have significant and different impacts on people’s behavioral preferences [16]. Under the definition of gain-loss framing effect, both “benefit” and “risk” are expressed as a subjective view and personal feeling of possible or assumed consequences. Specifically, health information with the gain frame will focus on defining the gains obtained by people from accepting a specific behavior. The information with loss frame, on the other hand, will highlight the risks associated with the rejection of such a health behavior [17-18]. Then, gain- and loss-framed messages have emerged as the essential tool to examine the framing effect in health communication studies [16-26].”

“Moreover, different types of health behaviors have also been introduced as moderators in analyzing the relationships between the framing effect and behavioral intentions [17-18]. Specifically, loss-framed messages have proven to be more persuasive in encouraging detection behaviors [19-21]; conversely, messages presented in gains are more persuasive in encouraging prevention behaviors [22-24]. Thus, gain framing may be more effective in promoting vaccination, which has been considered one type of prevention behavior [23-26]. However, regarding the global COVID-19 pandemic that continues to influence daily life significantly, it is unknown whether a gain-framed message will be more effective than a loss-framed message in promoting vaccination against COVID-19.”

  • At the end of the Introduction please report your hypotheses and not only your research questions.

Thanks very much for the comment. Covid-19 vaccine is newly developed and the impact of this pandemic is unprecedented. Therefore, this study is an exploratory study to explore the effect of message framing. This study is more application-focused. Therefore, we did not propose hypotheses. Instead, we only propose our research questions. 

  • How did you establish the sample size?

Thanks for your comments, and I added some descriptions about our sample size. We determine the sample size by following the recommendation of Simmons, Nelson, and Simonsohn’s study for online experiments. We also performed a sensitivity analysis and the results of sensitivity analysis showed that this sample size provided 80% power (α = .05) to detect small effects.

“College students were recruited from an online panel run by www.wjx.cn, which is like Amazon’s Mechanical Turk and Prolific. Students who have never got any COVID-19 vaccine were included in this study. Following Simmons, Nelson, and Simonsohn’s recommendations [85], we planned to target at least 50 per cell in online studies. As the final sample, 298 students (Mage = 21.44; 46.3% female; 63.7% undergraduate students and 36.3% graduate students (including 0.7% Ph.D. students) were recruited to participate in this study. The participants were invited to browse a web page, where they were shown a brief introduction and asked to sign an informed consent form.”

  • Regarding the development of the persuasive messages, did you pre-test it before the study? How did you establish the length of each message?

We did not pre-test the stimuli as the manipulation of gain-loss framing and narrative framing are well established. We determined the length of each message by following the normal length of health-related messages in China.

  • Please report the manipulation check questions in the section about measurement.

Thanks for your comments, we have added the manipulation checks questions to the section of measurement.

“The present study conducted a set of manipulation checks of the framed messages before measuring all related variables. The purpose of this is to test the effectiveness and accuracy of the message designs. The manipulation checks contained two dimensions of measurement and one is to measure how much participants think this is a piece of positive or negative message, and the other is to measure how much participants think this message is a narrative or non-narrative one. This study asked participants to complete the scoring of the following questions based on their feelings after reading the assigned health message, and the Likert scale ranging from 1 (strongly disagree) to 5 (strongly agree) was also used in the questionnaire. Specifically, six items included "This information emphasizes the benefits of getting COVID-19 vaccination"; "This information emphasizes the risk of not getting COVID-19 vaccination"; "This information emphasizes that getting COVID-19 vaccination will bring you positive effects"; "This information emphasizes that getting COVID- 19 vaccination will bring you negative effects"; "This information is written in a person's narrative"; "This information is written in an objective style".”

  • Please report the order of presentation of scales, this is very important given that you are testing also the mediation model.

Thanks for the comments. We made a brief description of the order of our measurements. We started with the measurement of manipulation checks and dependent variables. Next, it was the measurement of mediators, and it finally ended up with the demographic variable.

“Unless indicated otherwise, the responses to items were given on 5-point Likert scales. The Likert scale ranging from 1 (strongly disagree) to 5 (strongly agree) as end-points and the average mean values of related items were considered as indexes for the related variables. Before measuring all related variables, this study conducted the manipulation checks to examine the efficiency and reliability of the design of message stimuli. This study then measured the dependent variable, that is, intention to get the COVID-19 vaccine. Then, the present study continued to measure four main health beliefs as mediators, which include perceived severity, perceived susceptibility, perceived benefits, and perceived costs. Finally, this study measured demographical variables, including the participants' age, gender, education level, and Internet literacy.

  • The demographic features of the sample cannot be considered as results, please move this section to the design and participants section.

Thanks for your comments. We have moved this part to the design and participants sections.

  • In the description of Process MACRO for mediation analyses please report the bootstrap you have used (e.g., 1000, 2000, 5000). Further, explain to readers that the effect is significant whether the CIs do not include the zero value.

Thanks for your comments and we made some modifications in the results section.

“To address this question, an analysis of mediation was conducted by using the PROCESS macro in SPSS and the model 4 was selected. Using a bootstrap method, this study selected a sample size of 5000.”

“When testing the mediation effects of perceived severity, perceived benefits and perceived costs, their confidence intervals of the bootstrap did not include zero, which means that those three variables have significant mediating effects on narrative framing. Specifically, the mediating effects of perceived severity (LLCI = 0.0186, ULCI = 0.0868), perceived benefits (LLCI = 0.0137, ULCI = 0.1030), and perceived costs (LLCI = 0.0046, ULCI = 0.0660) were statistically significant in the 95% bias-corrected bootstrap confidence interval.”

  • In mediation analyses results the authors wrote “RQ4 emphasized that perceived susceptibility, perceived severity, perceived benefits, and perceived costs will have mediating roles on the relationship between narrative framing and vaccination intention” In my opinion this is a hypothesis, rather than a research question. Research question is to investigate whether the perceived susceptibility (and the other variables) would have a mediating role on the relationship between narrative framing and vaccination intention. The same for RQ5

Thanks for your comments. We made some modifications of the statements for the research questions.

“RQ4 tried to investigate whether and how narratives and non-narrative affect college students’ health beliefs, including perceived susceptibility, perceived severity, perceived benefits and perceived costs.”

“RQ5 then investigated whether the health beliefs mediate the relationship between narrative framing and intentions to get the vaccination. To address this research question, the coefficients of the mediating models were examined to analyze the directions of the mediating effects.”

  • A wide literature in persuasion have showed that individual differences in affective and cognitive orientation can predict the effectiveness of narrative (vs informative) messages (the so called matching effect, Mayer & Tormala, 2010, “Think” Versus “Feel” Framing Effects in Persuasion https://doi.org/10.1177/0146167210362981 Aquino et al., 2020, Sense or sensibility? The neuro-functional basis of the structural matching effect in persuasion, https://doi.org/10.3758/s13415-020-00784-7 ).

It could be interesting to investigate the role of affective and cognitive orientation in the field the authors are investigating. Please add this idea for future research (and the suggested references) in Discussion, so you can extend your manuscript to affective-cognitive persuasive field.

Thanks for your comments and recommendations. We have incorporated these two pieces of literature into the manuscript and discussed future research directions accordingly.

“Finally, the present study did not examine the roles of individual affective and cognitive orientations on promoting behavioral intentions. This study merely focused on analyzing the persuasiveness of messages with different expressions. However, people's affective and cognitive attitudes should also be introduced into the model. According to the theory of matching effect, if the framed message matches an individual's affective and cognitive orientations, it will enhance the effectiveness of persuasion [81]. In other words, matches between people's psychological states and the message will make it more persuasive [82-84]. Those related theories do give great inspiration for future research. According to the findings in this study, loss framing, compared with gain framing, is more persuasive. Additionally, narratives also have a more significant persuasive effect than non-narratives. However, this study did not take affective orientation and cognitive orientations into consideration. What type of framed messages can significantly match individual's psychological states and will better trigger their behavioral intentions? In future research, it may be necessary to add more variables to measure the dimensions of personal psychological characteristics to better measure the influence of emotion and cognition orientations on persuasiveness.”

Round 2

Reviewer 2 Report

I am happy about this revised version of the manuscript. I wish good luck to the authors for their future studies